# Prevalence and associating factors for iatrogenic gliosis-like changes in surgically treated intracranial meningioma patients—A retrospective study of 255 meningioma patients

**Joonas Laajava** *, Mika Niemelä, Miikka Korja

Department of Neurosurgery, University of Helsinki and Helsinki University Hospital, Helsinki, Finland

* Joonas.laajava@helsinki.fi

## Abstract

### Purpose

Persistent peritumoral brain edema (PTBE) following surgery of intracranial meningioma (IM) has recently been proposed to represent, in part, iatrogenic gliosis-like changes (IGCs). We aimed to estimate the frequency of IGCs after gross total resection (GTR) of IM, identify factors associated with IGCs, and assess their effect on surgical outcome.

### Methods

Patients with IM who underwent surgery between 2000 and 2020, presented with no preoperative PTBE on magnetic resonance imaging, and had at least one year of follow-up were retrospectively identified. Only patients without preoperative PTBE were included to ensure that postoperative hyperintense fluid-attenuated inversion recovery (FLAIR) changes were iatrogenic. Outcomes were evaluated based on postoperative symptoms and changes in Karnofsky Performance Status (KPS).

### Results

A total of 255 meningioma patients without preoperative PTBE were identified. Of these, 133 (52.2%) showed postoperative IGCs on FLAIR imaging. IM location (p < .001), volume (p < .001) and World Health Organization grade (p = .003) associated with IGCs in univariate analysis. In multivariate analysis, non-convexity location (OR 2.0–12.2, p = .04 to <.001) and IM volume (OR 1.1, p < .001) remained significant. Patients with IGCs showed a modest KPS improvement from 86.7 to 88.0 (+1.3), while those without IGCs improved more from 90.0 to 92.5 (+2.5, p = .002). IGCs were associated with residual symptoms (OR 2.3, p = .01) and new-onset seizures (OR 6.1, p = .04).

**Data availability statement:** This study was approved by the Ethics Board of Helsinki University Hospital (HUH). Under Finnish legislation, register-based studies do not require individual patient consent. Due to legal data-transfer restrictions imposed by the European Union, the de-identified dataset used in this study cannot be shared directly. Requests for access to the original data may be submitted to the Finnish Health and Social Data Permit Authority (Findata): http://findata.fi/en/.

**Funding:** The author(s) received no specific funding for this work.

**Competing interests:** The authors have declared that no competing interests exist.

## Conclusion

IGCs are frequent following GTR of IM, and associate with IM location and volume. IGCs likely associate with impaired recovery following surgery and new-onset seizures.

## Introduction

Intracranial meningiomas (IMs) constitute the most prevalent form of intracranial tumors [1] and are classified into three grades by the World Health Organization (WHO) [2]. Typically, meningiomas are identified and monitored with magnetic resonance imaging (MRI) [3]. The current standard of care for meningiomas includes observation or surgical resection, often complemented by radiotherapy, particularly in cases of atypical and anaplastic meningiomas [4,5]. Meningiomas frequently exhibit peritumoral brain edema (PTBE) [6–8]. Recent literature suggests that persistent PTBE may partly represent gliosis [9–11], which can sometimes be iatrogenic. Differentiating between persistent PTBE and gliosis remains challenging due to their similar appearance on MRI sequences [11,12].

Preoperative PTBE has been linked with pre- and postoperative seizures [13–15]. Additionally, a recent study suggests that postoperative T2/FLAIR hyperintense MRI changes may contribute to postoperative seizures [9]. Although iatrogenic changes from surgical intervention have been proposed as a contributor to persisting PTBE, no studies have systematically evaluated the prevalence of iatrogenic gliosis- like changes (IGCs) following IM surgery.

To investigate the prevalence and significance of IGCs, we assessed the occurrence of persistent FLAIR hyperintensity changes in IM patients without preoperative PTBE who underwent gross total resection (GTR). Given that PTBE frequently persists [11], assessing the prevalence of iatrogenic changes in patients with preoperative PTBE is unreliable. Moreover, assessment should be conducted no earlier than one year postoperatively to allow sufficient time for any transient changes to resolve [9]. By focusing exclusively on patients who underwent GTR and lacked preoperative PTBE or other hyperintense peritumoral findings, we aimed to ensure that any newly observed changes were iatrogenic in origin. Our aim was to estimate the prevalence of persistent IGCs and to explore whether the presence of persistent IGCs is associated with surgical outcome.

## Methods

### Ethics statements

This study was approved in 2023 by the local Institutional Research Board. Patient data was pseudonymized. As patients could not be identified, the need for consent was waived. Due to local data privacy laws, data cannot be exported from the local cloud-based cybersecure operating environment, which has been designed for storing, processing and analyzing research data. The data was accessed for study purposes between 20/12/2023 and 16/02/2025. The study was conducted in accordance with the Helsinki Declaration.

## Study population

All adult patients who underwent GTR of supratentorial IM at the study hospital between 2000 and 2020 were retrospectively identified. Details on patient data acquisition have been described in a previous publication [11]. Briefly, patients were identified using international Classification of Diseases 10 diagnosis code D32 (benign neoplasm of cerebral meninges) and data was obtained from electronic patient records (Uranus, Opera, RADU, picture archiving and communication system, Qpati).

## Inclusion criteria

The inclusion criteria were: (1) GTR of histologically confirmed supratentorial IM; (2) first intracranial surgery; (3) absence of preoperative PTBE or other hyperintense peritumoral findings in the most recent MRI scan prior to surgery; (4) surgery performed within one year of the preoperative MRI; (5) availability of postoperative MRI with a minimum follow-up time of one year; (6) MRI quality of at least a 1.5T scanner, including contrast-enhanced images and FLAIR sequences; (7) absence of other intracranial tumors; (8) no previous intracranial radiotherapy to exclude radiation-induced gliosis [16]; (9) absence of recurrent or residual tumors; and (10) availability of clinical outcome data for two years postoperatively.

## Iatrogenic gliosis-like changes

IGCs were evaluated using postoperative MRI scans acquired a minimum of one year after surgery, ensuring sufficient time for transient changes to resolve [9]. The study hospital's follow-up protocol has been described previously [11]. Briefly, the first postoperative MRI for WHO grade 1 IMs was usually performed at two years, while the first postoperative MRI for WHO grade 2 and skull base IMs was usually performed at one year postoperatively. Any T2/FLAIR hyperintensity surrounding the surgical cavity detected in the first postoperative MRI (a minimum of one year following surgery) was classified as an IGC. We considered no acceptable peritumoral rim of hyperintensity. All surgeries were performed using either a microscope or an exoscope.

We assessed whether WHO grade, sex, age, IM location, IM volume and T2 signal intensity of IM were associated with IGCs. Patients were categorized into three age groups: young adults (18–39), middle-aged adults (40–64), and older adults (65+). Detailed measurements of IM parameters and postoperative IGCs were based on manual segmentations on MR images [17]. The segmentation methods have been described previously in detail [11]. We looked at the T2 signal intensity of IMs, as T2 signal intensity has been proposed to indicate IM firmness [18]. We classified IMs as hyperintense if more than 75% of the T2 signal was hyperintense, and as hypointense if more than 75% of the T2 signal was hypointense. IM locations were categorized into five groups: convexity, falx, parasagittal, skull base and intraventricular/tentorial [19]. Anterior clinoid, olfactory groove, planum sphenoidale and sphenoid wing IMs were classified as skull base IMs. IGC location was assigned to the most affected brain lobule: frontal, occipital, parietal or temporal.

## Surgical outcome

All patients were followed up for a minimum of two years through electronic patient records (Uranus, CGI, Montreal, Quebec, Canada). These records included all visits to specialized medical care across hospitals operated by the Hospital District of Helsinki and Uusimaa (HUS). Patients whose follow-up occurred outside HUS area were excluded due to insufficient follow-up data.

Surgical outcomes were evaluated using the Karnofsky Performance Status (KPS) assessment [20], based on patient records. All KPS scores were assessed by a single researcher (JL) using predetermined criteria to ensure consistency and comparability across patients. The presence of preoperative and postoperative symptoms was also evaluated. A patient was considered symptomatic if head imaging was performed due to a presenting symptom and if a neurologist or neurosurgeon considered that the symptom related to the IM. Postoperatively, a patient was deemed symptomatic if

a neurosurgeon or neurologist recorded a new symptom or deficit that associating with the surgical procedure. Patients were categorized as having new-onset symptoms if no preoperative symptoms were identified and as having residual symptoms if some symptoms remained postoperatively.

Patients who experienced at least one non-early (>1 week following surgery) onset seizure during follow-up were classified as having postoperative seizures. Non-early postoperative seizures, we divided them into persisting and new-onset seizures. Early-onset seizures were defined as those occurring within the first postoperative week [21].

### Statistical methods

We applied Chi-squared test in assessing the association between categorical and binary outcome variables. For continuous variables, we used the Mann-Whitney U test (Wilcoxon rank-sum test) to comparing differences. Normality was assessed using the Shapiro-Wilk test. Age, as the only normally distributed continuous variable, was assessed with the Student's t-test. We used a binary logistic regression model in multivariate analyses. Multivariate analyses with 95% confidence intervals included variables that were identified significant (p < .05) in univariate analyses. For categorical variables, odds ratios (ORs) represented the odds in comparison to the reference category. For continuous variables, ORs represented the change in odds associated with a one-unit increase in the variable. For IM location and WHO grade, the reference category was the one with most occurrences. IM volume was modeled as a linear variable, and coefficients were interpreted as the change in log-odds of the outcome for each one-unit increase in the IM volume. We considered p-values of <.05 significant. All statistical analyses were performed in R (R Core Team, 2023) [22].

## Results

### Study cohort

Complete data was available for all study patients. Of the 5480 IM patients diagnosed between 2000 and 2020, we identified 255 IM patients (Table 1) who a) showed no PTBE in preoperative MRIs, b) underwent GTR, c) had pre- and postoperative MRI images available, d) were followed-up with MRI for a minimum of one year and e) had comprehensive clinical patient records available (Fig 1).

### Iatrogenic lesions

Of the 255 operated IM patients, 133 (52.2%) showed postoperative FLAIR hyperintensity in the first available postoperative MRI after a minimum of one year (median 2.2 years, range 1.0–6.6 years) of follow-up (Table 1). The incidence of IGCs varied from 26.8% to 83.6% by IM volume (Fig 2). Postoperative IGC volume ranged from 0.05 cm$^3$ to 31.6 cm$^3$ in patients with IGCs. IGCs were most infrequent in patients with convexity IMs, and most frequent in patients with intraventricular/tentorial IMs (Fig 3).

In univariate analysis, IGCs associated with IM location (p < .001), increasing IM volume (p < .001) and WHO grade 2 (p = .003) (Table 2). In multivariate analysis, skull base (OR 2.0, p = .04), parasagittal (OR 8.1, p < .001), falcine (OR 3.5, p = .01) or intraventricular/tentorial (12.2, p = .03) locations, as well as increasing IM volume (OR 1.1 per 1 cm$^3$, p < .001) associated with the incidence of postoperative IGCs (Table 2).

None of the WHO grade 2 meningiomas, nor any other meningiomas in the cohort, received radiation before or during the study's observation period, ensuring that no imaging changes could be attributed to irradiation.

### Surgical outcome

Patients with postoperative IGCs had a mean preoperative KPS of 86.7, which improved to 88.0 postoperatively—a 1.3-point increase—over the two-year clinical follow-up. In contrast, patients without postoperative IGCs had a higher mean preoperative KPS of 90.0 and demonstrated a greater improvement of 2.5 points, reaching 92.5 postoperatively (p = .002,

**Table 1. Patient characteristics.**

|  | Overall |
|---|---|
| Overall, n (%) | 255 (100%) |
| Age in years, median (IR) | 54.0 (45.5-63.0) |
| Age group, n (%) |  |
| Young adult (18–39) | 28 (11.0%) |
| Middle aged adult (40–64) | 173 (67.8%) |
| Older adult (65+) | 54 (21.2%) |
| Sex, n (%) |  |
| Women | 200 (78.4%) |
| Men | 55 (21.6%) |
| IM location, n (%) |  |
| Convexity | 95 (37.3%) |
| Skull base | 86 (33.7%) |
| Parasagittal | 33 (12.9%) |
| Falx | 32 (12.5%) |
| Intraventricular/Tentorial | 9 (3.5%) |
| IM laterality, n (%) |  |
| Left | 118 (46.3%) |
| Right | 96 (37.6%) |
| Bilateral | 41 (16.1%) |
| WHO Grade, n (%) |  |
| 1 | 237 (92.9%) |
| 2 | 18 (7.1%) |
| IM volume in cm$^3$, median (IR) | 3.01 (1.2-9.1) |
| Preoperative IM T2-intensity |  |
| Isointense | 162 (63.5%) |
| Hyperintense | 82 (32.2%) |
| Hypointense | 11 (4.3%) |
| Preoperative KPS, mean (SD) | 88.3 (11.7) |
| Postoperative KPS at 2 years, mean (SD) | 90.2 (12.7) |
| Preoperative seizures, n (%) |  |
| No | 223 (87.5%) |
| Yes | 32 (12.5%) |
| Postoperative seizure during 2 years, n (%) |  |
| No | 238 (93.3%) |
| Yes | 17 (6.7%) |
| Time to postoperative MRI in years, median (IR) | 2.2 (1.6-2.6) |
| Postoperative IGC volume, median (IR) | 0.1 (0.0-0.8) |

IGC = Iatrogenic gliosis-like change, IM = Intracranial meningioma, IR = Interquartile range, KPS = Karnofsky performance status, MRI = Magnetic resonance imaging, SD = Standard deviation, WHO = World Health Organization.

Table 3). While this difference in the composite score is statistically significant, it is not clinically meaningful and may be explained by measurement variability. However, a clinically significant worsening of KPS (defined as a decline of ≥10 points) was observed in 23 of 133 patients (17.3%) with IGCs, compared to only nine of 122 patients (7.3%) without IGCs

## Patient selection process

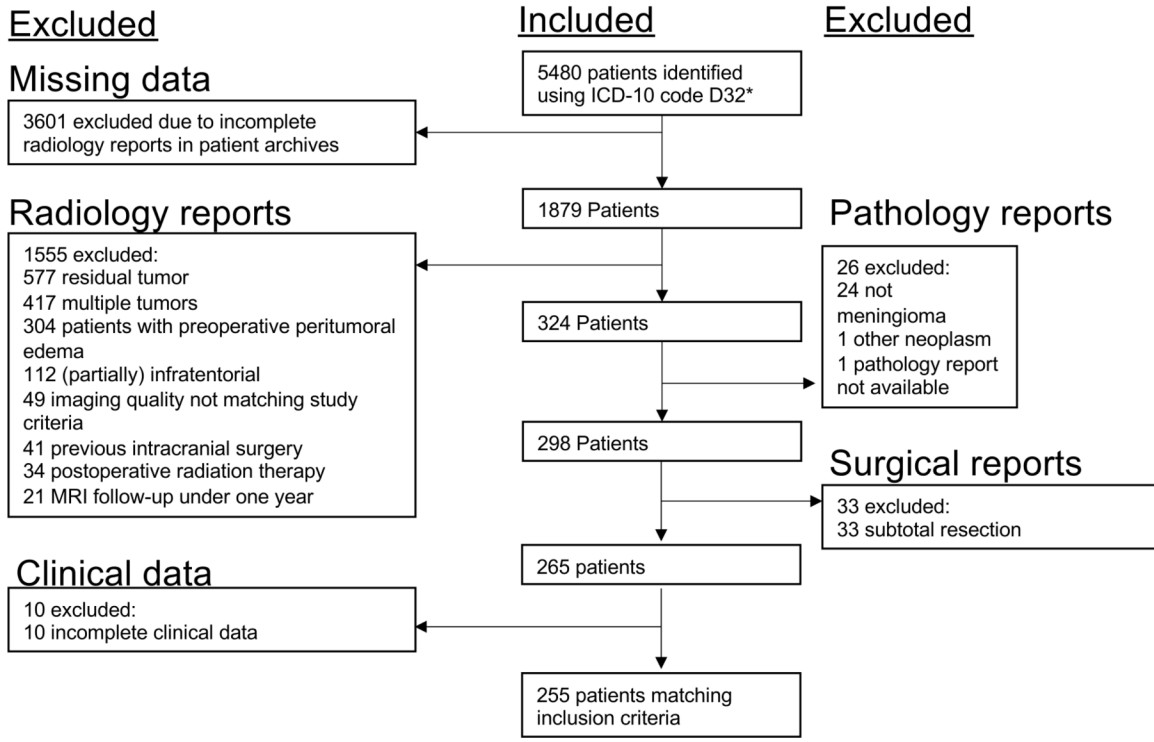

**Fig 1. Study selection process.** ICD = International classification of diseases, MRI = Magnetic resonance imaging.

(p = .03). Additionally, all cases with considerable functional impairment (KPS ≤ 50) occurred in the IGC group, affecting five patients (3.8%) out of 133. These findings suggest that the presence of IGCs is possibly associated with clinically relevant functional deterioration in a subset of patients.

Preoperative symptoms were more common in patients with postoperative IGCs (p = .04, OR 1.7, Table 3). Patients with IGCs were more likely to have residual symptoms postoperatively (p = .01, OR 2.3, Table 3). Preoperative seizures were more common in patients with postoperative IGCs (p = .04, OR 2.2, Table 3). Again, this is likely due to differences in preoperative IM volumes and WHO grade 2 IMs (Table 2). IGCs were associated with new-onset seizures (p = .04, OR 6.1, Table 3).

## Discussion

IGCs are frequent following GTR of IM, particularly when using the described strict definition of IGCs. IGCs associated with non-convexity IM location and IM volume. The presence of IGCs also associated with residual symptoms as well as new-onset seizures. In line with our hypothesis, the presence of IGCs impaired the functional postoperative improvement, when assessed using the KPS score.

Using strict criteria for PTBE-like changes, these changes have been suggested to persist postoperatively in nearly all cases [11]. We have previously speculated that IGCs could explain up to 50% of persisting PTBE [11]. The current study findings seem to confirm our speculation, as 52.2% of operated IM patients had IGCs. No previous studies have looked at prevalence of IGCs in patients with no preoperative PTBE or other hyperintense peritumoral findings. The likelihood of

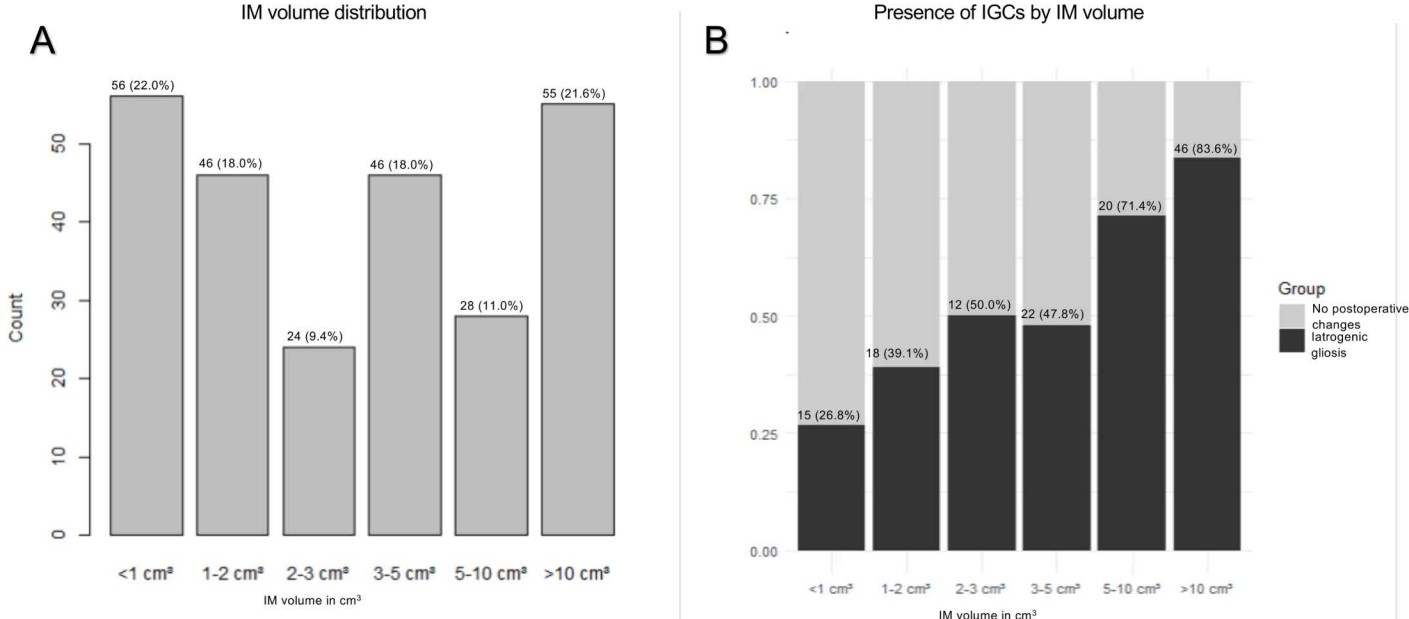

**Fig 2. Iatrogenic gliosis-like changes by meningioma volume.** (A) Histogram showing the distribution of intracranial meningioma (IM) volume in the study cohort. Numbers above bars represent patient counts. (B) Histogram showing prevalence of iatrogenic gliosis-like changes (IGCs) by IM volume. Numbers above bars represent patient counts.

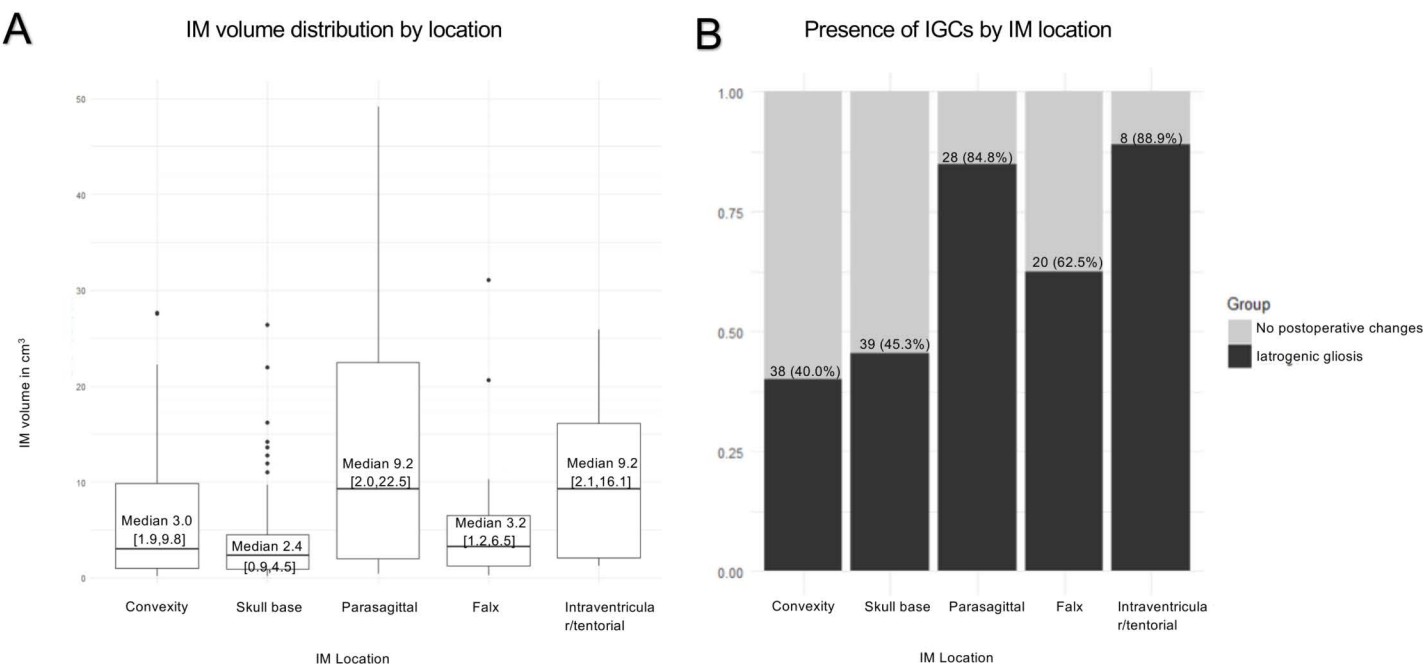

**Fig 3. Iatrogenic gliosis-like changes by meningioma location.** (A) box and whiskers plot presenting intracranial meningioma (IM) volume by location. Median volumes with interquartile ranges are presented inside boxes. (B) Histogram presenting the incidence of iatrogenic gliosis-like changes (IGCs) by IM location. Numbers above bars represent patient counts.

**Table 2. Univariate and Multivariate Analyses of Factors Associating with IGCs.**

| | No postoperative gliosis-like changes present (n = 122) | Iatrogenic gliosis-like changes present (n = 133) | Univariate P value | Multivariate/OR P value |
|---|---|---|---|---|
| Age in years, median (IR) | 54.0 (44.0-63.0) | 56.0 (47.0-64.0) | .49 [a] | |
| Age group, n (%) | | | | |
| Young adult (18–39) | 16 (57.1%) | 12 (42.9%) | .54[b] | |
| Middle aged adult (40–64) | 82 (47.4%) | 91 (52.6%) | | |
| Older adult (65+) | 24 (44.4%) | 30 (55.6%) | | |
| Sex, n (%) | | | | |
| Women | 99 (49.0%) | 101 (51.0%) | .31 [b] | |
| Men | 23 (41.8%) | 32 (58.2%) | | |
| IM location, n (%) | | | | |
| Convexity | 57 (60.0%) | 38 (40.0%) | <0.001[b] | |
| Skull base | 47 (54.7%) | 39 (45.3%) | | 0.04 (OR: 2.0) 95% CI [1.0,3.9] |
| Parasagittal | 5 (15.2%) | 28 (84.8%) | | <0.001 (OR: 8.1) 95% CI [2.8,27.3] |
| Falx | 12 (37.5%) | 20 (62.5%) | | 0.0057 (OR: 3.5) 95% CI [1.5,8.8] |
| Intraventricular/Tentorial | 1 (11.1%) | 8 (88.9%) | | 0.026 (OR: 12.2) 95% CI [1.9,240.8] |
| **Convexity** | | | | |
| Convexity (R) | 57 (60.0%) | 38 (40.0%) | .003[b] | N/A |
| Non-convexity | 65 (40.6%) | 95 (59.4%) | (OR: 2.2) | |
| IM laterality, n (%) | | | | |
| Left | 52 (44.1%) | 66 (55.9%) | .40 [b] | |
| Right | 47 (49.0%) | 49 (51.0%) | | |
| Bilateral | 23 (56.1%) | 18 (43.9%) | | |
| WHO Grade, n (%) | | | | |
| 1 | 120 (50.6%) | 117 (49.4%) | .003 [b] | |
| 2 | 2 (11.1%) | 16 (88.9%) | | 0.42 (OR: 2.1) 95% CI [0.4,15.7] |
| IM volume in cm³, median (IR) | 1.52 (0.9-3.5) | 4.85 (2.1-16.2) | <0.001 [c] | <0.001 (OR: 1.1) 95% CI [1.1,1.2] |
| Preoperative IM T2-intensity | | | | |
| Isointense | 80 (49.4%) | 82 (50.6%) | .12 [b] | |
| Hyperintense | 34 (41.5%) | 48 (58.5%) | | |
| Hypointense | 8 (72.7%) | 3 (27.3%) | | |
| Time to postoperative MRI in years, median (IR) | 2.2 (1.6-2.5) | 2.2 (1.7-2.8) | .66 [c] | |

CI = Confidence interval, IGC = Iatrogenic gliosis-like change, IM = Intracranial meningioma, IR = Interquartile range, KPS = Karnofsky performance status, MRI = Magnetic resonance imaging, OR = Odds ratio, SD = Standard deviation, WHO = World Health Organization.

[a] Student's t-test.

[b] Chi-squared test.

[c] Mann-Whitney U test.

**Table 3. Postoperative symptoms in relation to IGCs.**

| | No postoperative gliosis-like changes present (n = 122) | Iatrogenic gliosis-like changes present (n = 133) | Univariate/OR p value |
|---|---|---|---|
| Preoperative KPS, mean (SD) | 90 (10.5) | 86.7 (12.6) | .05[a] |
| Postoperative KPS at two years, mean (SD) | 92.5 (11.2) | 88.0 (13.6) | .002[a] |
| Preoperative symptoms, n (%) | | | |
| No | 83 (52.9%) | 74 (47.1%) | |
| Yes | 39 (39.8%) | 59 (60.2%) | .04[b] (OR: 1.7) 95% CI [1.0,2.8] |
| Postoperative symptoms, n (%) | | | |
| No | 100 (53.2%) | 88 (46.8%) | |
| Residual | 16 (33.3%) | 32 (66.7%) | .01[b] (OR: 2.3) 95% CI [1.2,4.5] |
| New-Onset | 6 (31.6%) | 13 (68.4%) | .07[b] (OR: 2.4) 95% CI [0.9,7.3] |
| Preoperative seizure, n (%) | | | |
| No | 112 (50.2%) | 111 (49.8%) | |
| Yes | 10 (31.3%) | 22 (68.8%) | .04[b] (OR: 2.2) 95% CI [1.0,5.1] |
| Postoperative seizure during two years, n (%) | | | |
| No | 118 (49.6%) | 120 (50.4%) | |
| Persisting | 3 (33.3%) | 6 (66.7%) | .34[b] (OR: 1.9) 95% CI [0.5,9.8] |
| New-Onset | 1 (12.5%) | 7 (87.5%) | .04[b] (OR: 6.1) 95% CI [1.0,157.5] |

KPS = Karnofsky performance status, OR = Odds ratio, SD = Standard deviation.

[a] Mann Whitney U Test.

[b] Chi-squared test.

IGCs appears to increase with increasing IM volumes. Therefore, depending on the IM cohort characteristics and probably also on surgical techniques, the incidence of IGCs is likely highly variable. In terms of surgical techniques, whether an extensive debulking of high-volume IMs to minimize the size before removal could decrease the incidence of IGCs may be worth studying in future. Moreover, a water dissection technique could be useful in minimizing ICGs [23], but also this needs to be prospectively studied.

Large academic teaching hospitals could define both overall and surgeon-specific outcomes. According to our previous studies, high-quality academic training hospitals demonstrate that surgeons achieve comparable results, regardless of their individual experience levels [24,25]. Given that complications are rare in such high-quality settings, subtle differences in surgical outcomes may be more effectively assessed using MRI, which offers greater sensitivity to detecting postoperative changes.

Gliosis may develop secondary to venous infarction [26]. Therefore, anatomical location may expose, for example, parasagittal meningioma patients to a higher risk of IGCs. Another mechanism causing IGCs is brain tissue manipulation, such as retraction, which can induce microglial activation and inflammation, leading to gliosis [27–30]. Indeed, previous studies have suggested that patients with parasagittal [31–33], falcine [33,34], skull base [35], tentorial [36,37] and intraventricular [38] IMs have an increased risk of venous infarctions and manipulation-related injuries.

As IGCs represent iatrogenic damage to brain tissue, it is perhaps not surprising that we observed impaired postoperative functional improvement as well as a higher prevalence of residual symptoms and new-onset seizures in patients with IGCs. In fact, seven out of eight cases of new-onset postoperative seizures occurred in patients with postoperative IGCs. All IGCs in these seven patients with new-onset seizures located in the frontal lobes (S1 Table). Of these seven patients, four (57.1%) had relatively extensive subcortical lesions, while three (42.9%) had smaller cortical lesions.

In mice studies, gliosis-associated seizures have been speculated to be caused by adenosine kinase overexpression [39,40], which may lower seizure threshold by reduction of extracellular adenosine [41]. Furthermore, brain injury and

gliosis have been suggested to cause disruptions to glutamate metabolism leading to increased susceptibility to seizures in mice [42,43]. Disruptions to glutamate metabolism have also been shown in humans following traumatic brain injury [44]. Lastly, in one mice study, gliosis is thought to potentially lower seizure threshold through loss of GABA in inhibitory neurons [45]. Considering these mechanisms, it is perhaps expected that IGCs are associated with new-onset postoperative seizures. However, to our knowledge, the association between IGCs and new-onset postoperative seizures has not been previously reported in humans.

### Limitations

Our study has a few notable limitations. First, surgical outcomes were assessed based on a retrospective review of patient records, and patients did not undergo a systematic follow-up protocol. Therefore, it is possible that some functional and particularly neurocognitive symptoms related to surgery were missed or underreported. Quality of life data was also not available. Second, we did not have detailed data on the use of antiepileptic medication. It is likely that the postoperative use of antiepileptic medication affects the incidence of postoperative seizures. Therefore, the incidence of postoperative seizures in our study does not reflect the true postoperative incidence, depending on whether all operated patients were or were not on antiepileptic medication. However, this limitation is unlikely to change the study conclusion suggesting that IGCs associate with new-onset seizures.

Additionally, due to the retrospective nature of this study, determining the exact etiology of IGC changes is not possible. However, our goal was not to establish causation, but rather to assess how commonly these changes occur. The observed IGCs likely reflect a heterogenous set of underlying mechanisms, which cannot be definitively distinguished in this study design.

We did not perform multivariate analysis for postoperative symptoms, as such modeling would not be meaningful in this context. Our objective was not to account for all potential contributors to these symptoms, but rather to isolate the association between the presence of IGCs and specific outcomes. Multivariate adjustment would not alter the key finding that IGCs are associated with certain postoperative symptoms, such as new-onset seizures.

### Conclusions

IGCs are common following GTR of IMs and occur more frequently after GTR of large and of non-convexity IMs. IGCs appear to decrease the likelihood of functional improvement after surgery and likely increase the risk of new-onset seizures. Whether IGCs in IM patients can act as surgically resectable epileptogenic foci may be worth studying in the future.

### Supporting information

**S1 Table. Characteristics of patients with postoperative seizures.** IGC = Iatrogenic gliosis-like change, IM = Intracranial meningioma, IR = Interquartile range, KPS = Karnofsky performance status, MRI = Magnetic resonance imaging, SD = Standard deviation, WHO = World Health Organization.
(DOCX)

### Author contributions

**Conceptualization:** Joonas Laajava, Miikka Korja.

**Data curation:** Joonas Laajava.

**Formal analysis:** Joonas Laajava.

**Funding acquisition:** Joonas Laajava.

**Investigation:** Joonas Laajava.

**Methodology:** Joonas Laajava, Miikka Korja.

**Project administration:** Mika Niemelä, Miikka Korja.

**Resources:** Mika Niemelä, Miikka Korja.

**Supervision:** Mika Niemelä, Miikka Korja.

**Validation:** Mika Niemelä, Miikka Korja.

**Writing – original draft:** Joonas Laajava.

**Writing – review & editing:** Mika Niemelä, Miikka Korja.

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
