## [Decision Letter · Decision Letter 0]

9 Nov 2025

Dear Dr. Mikael,

Thank you for submitting your manuscript to PLOS ONE. After careful consideration, we feel that it has merit but does not fully meet PLOS ONE’s publication criteria as it currently stands. Therefore, we invite you to submit a revised version of the manuscript that addresses the points raised during the review process.

We look forward to receiving your revised manuscript.

Kind regards,

Ryota Tamura

Academic Editor

PLOS ONE

Journal Requirements:

Additional Editor Comments:

Minor revision is needed. Thank you very much for your effort.

Reviewers' comments:

Reviewer's Responses to Questions

**Comments to the Author**

1. Is the manuscript technically sound, and do the data support the conclusions?

Reviewer #1: Yes

Reviewer #2: Yes

Reviewer #3: Yes

Reviewer #4: Yes

2. Has the statistical analysis been performed appropriately and rigorously?

Reviewer #1: Yes

Reviewer #2: Yes

Reviewer #3: Yes

Reviewer #4: Yes

3. Have the authors made all data underlying the findings in their manuscript fully available?

Reviewer #1: Yes

Reviewer #2: Yes

Reviewer #3: No

Reviewer #4: No

4. Is the manuscript presented in an intelligible fashion and written in standard English?

Reviewer #1: Yes

Reviewer #2: Yes

Reviewer #3: Yes

Reviewer #4: Yes

Reviewer #1: Dear Author,

I read and reviewed your article titled "Prevalence and Associated Factors for Iatrogenic Gliosis-Like Changes in Surgically Treated Intracranial Meningioma Patients – A Retrospective Study of 255 Meningioma Patients" with interest. This is a study that will guide the literature. Thank you.

Reviewer #2: Dear Author,

I had the opportunity to review your study titled "Prevalence and Associated Factors for Iatrogenic Gliosis-Like Changes in Surgically Treated Intracranial Meningioma Patients – A Retrospective Study of 255 Meningioma Patients."

I believe your results, which are consistent with the literature, will be beneficial in the scientific arena.

Thank you.

Reviewer #3: This is a follow-up manuscript from a single center retrospective cohort of surgically treated patients with meningioma that aimed to measure the odds of developing IGC as diagnosed with follow-up MRI. Authors conclude that the presence of IGC was associated with GTR of large and non-convexity intracranial meningiomas. The article is well presented and easy to follow.

Minor areas of opportunity are:

1: In table 1 and 2 please consider changing WHO Grade from roman numbers to arabic. (1 instead of I, and 2 instead of II).

2: Self-citation is not encouraged, see references 9 and 11.

3: As this was an exploratory study, hypothesis could be obviated; nevertheless, authors wrote two hypothesis, these hypothesis are usually used to calculate the sample size (which was not done) considering the direction and magnitude of the null hypothesis. Reconsider and follow proper research methods

4: Was RANO, QLQ-BN20 considered at any time?

5: It is easier to decribe the P value as <0.01 rather than the negative exponential (Table 2); and the 95%CI is almost never the same value as the OR (i.e., OR 1.1; 95% CI, 1.1, 1.2). OR for non-convexity vs convexity should also be included for one of the conclusions is precisely this. This last comment should is also to be considered for GTR vs non-GTR.

6: Large volume cut-off value should be included or the phrase large should change (suggestion: presence of IGC was associated with GTR (place OR, 95% CI, and P values), volume (place OR, 95%CI, and P values), and non-convexity (place OR, 95%CI, and P value).

Congratulations on your line of reasearch

Reviewer #4: The authors retrospectively reviewed their series of 5480 patients with intracranial meningioma, treated between 200o and 2020, and included 225 patients who had no preoperative peritumoral brain edema, underwent total resection of the meningioma, and in whom both pre-operative MRI and MRI at least 1 year post-surgery was available.

They assessed the frequency of iatrogenic gloss like changes (IGC), which they defined as any T2/FLAIR hyperintensity surrounding the surgical cavity.

IGC occurred among 52%, and was associated with non-convexity tumor location, and pre-operative tumor volume.

The methods used are sound and the conclusion is based on the presented data. The paper is well written and very readable.

The median time to first MRI follow-up is 2,2 years.

I have only 1 question:

Patients who underwent previous radiation therapy were excluded. However in 18 patients a WHO grade 2 meningioma was diagnosed. Did they undergo radiation treatment only after the first MRI at least one year later?

And one textual remark: On p3, line 87/88, this sentence may improve when the authors write: ..., we assessed the occurrence of persistent FLAIR hyper intensity changes in IM patients without preoperative PTBE, who underwent gross total resection (GTR).

**Do you want your identity to be public for this peer review?** For information about this choice, including consent withdrawal, please see our Privacy Policy

Reviewer #1: **Yes: ** Mehmet Edip Akyol

Reviewer #2: **Yes: ** Özkan Arabacı

Reviewer #3: **Yes: ** Bernardo Cacho-Díaz, MD, MSc, PhD

Reviewer #4: **Yes: ** Dennis R. Buis

---

## [Author Response · Author response to Decision Letter 1]

11 Nov 2025

Department of Neurosurgery

Submission ID: PONE-D-25-39472

MS title: ” Prevalence and associating factors for iatrogenic gliosis-like changes in surgically treated intracranial meningioma patients – A retrospective study of 255 meningioma patients”

Dear Dr Ryota Tamura,

We would like to express our gratitude for your consideration and the valuable feedback provided by the referees regarding our manuscript. We are grateful for the opportunity to submit a revised version. The reviewers have brought up important aspects that were all taken into account when revising the manuscript. In other words, we have made every effort to address all the comments and questions. Below, please find our detailed responses to the editor and reviewers. Thank you very much.

On behalf of all authors,

Editor

Thank you for highlighting the PLOS ONE style requirements, including the guidelines for file naming. We have revised and reformatted the manuscript to ensure full compliance with the PLOS ONE formatting standards and sample templates.

2.

This study was approved by the Ethics Board of Helsinki University Hospital (HUH). Under Finnish legislation, register-based studies do not require individual patient consent. Due to legal data-transfer restrictions imposed by the European Union, the de-identified dataset used in this study cannot be shared directly. Requests for access to the original data may be submitted to the Finnish Health and Social Data Permit Authority (Findata): http://findata.fi/en/.

Thank you for the clarification. The reviewers did not request the citation of any specific previously published works.

Thank you for the guidance. We have reviewed the entire reference list and confirmed that all entries are complete and accurate. None of the cited works have been retracted. No changes to the reference list were required.

Reviewer #1

Dear Author,

I read and reviewed your article titled "Prevalence and Associated Factors for Iatrogenic Gliosis-Like Changes in Surgically Treated Intracranial Meningioma Patients – A Retrospective Study of 255 Meningioma Patients" with interest. This is a study that will guide the literature. Thank you.

We sincerely thank the Reviewer for the delightful feedback.

Reviewer #2

Dear Author,

I had the opportunity to review your study titled "Prevalence and Associated Factors for Iatrogenic Gliosis-Like Changes in Surgically Treated Intracranial Meningioma Patients – A Retrospective Study of 255 Meningioma Patients."

I believe your results, which are consistent with the literature, will be beneficial in the scientific arena.

Thank you.

We are grateful to the Reviewer for the time invested in reviewing our manuscript.

Reviewer #3:

This is a follow-up manuscript from a single center retrospective cohort of surgically treated patients with meningioma that aimed to measure the odds of developing IGC as diagnosed with follow-up MRI. Authors conclude that the presence of IGC was associated with GTR of large and non-convexity intracranial meningiomas. The article is well presented and easy to follow.

Minor areas of opportunity are:

1: In table 1 and 2 please consider changing WHO Grade from roman numbers to arabic. (1 instead of I, and 2 instead of II).

We thank the Reviewer for this relevant and appropriate comment. Accordingly, we have made the following changes to the manuscript:

Table 1 and Table 2: WHO grading changed to Arabic numerals.

2: Self-citation is not encouraged, see references 9 and 11.

Thank you very much for this excellent remark. We fully agree that self-citations should be avoided. In this case, however, we hope that the citations are appropriate, as the cited findings and facts are available only in our earlier publications.

3: As this was an exploratory study, hypothesis could be obviated; nevertheless, authors wrote two hypothesis, these hypothesis are usually used to calculate the sample size (which was not done) considering the direction and magnitude of the null hypothesis. Reconsider and follow proper research methods

We again thank the reviewer for the professional comments. We have made the following changes to the manuscript:

Under abstract, Manuscript file page 2 line 43-44: We hypothesized that ICGs are infrequent but impact outcomes.

Under introduction, Manuscript file page 3 line 93-95: We hypothesized that persistent IGCs would occur infrequently. Furthermore, we hypothesized that IGCs associate with impaired surgical outcomes. � Our aim was to estimate the prevalence of persistent IGCs and to explore whether the presence of persistent IGCs is associated with surgical outcome.

Under discussion, Manuscript file page 10 line 262: Contrary to our hypothesis,

4: Was RANO, QLQ-BN20 considered at any time?

We applied RANO principles when assessing postoperative imaging. All meningioma measurements were performed on T1-weighted MRI, and in this cohort every tumor was completely resected with no recurrences observed. While a full RANO-based reporting framework is generally applied in clinical trials, our study was observational and not trial-based, so comprehensive RANO reporting was not performed.

We agree that QLQ-BN20 data would add valuable insight. Unfortunately, quality-of-life measures are not routinely collected at our institution, and these data were therefore unavailable for this study. We have added the following statement to the limitations section:

Under limitations, Manuscipr file page 12 line 313: “Quality of life data was also not available”

5: It is easier to decribe the P value as <0.01 rather than the negative exponential (Table 2); and the 95%CI is almost never the same value as the OR (i.e., OR 1.1; 95% CI, 1.1, 1.2). OR for non-convexity vs convexity should also be included for one of the conclusions is precisely this. This last comment should is also to be considered for GTR vs non-GTR.

We thank the Reviewer for the comments that surely improve the manuscript. We absolutely agree that reporting p-values without negative exponents improves clarity, and therefore we have updated Table 2 accordingly. We also agree that a convexity versus non-convexity analysis is informative. Unfortuntaly, as all included meningiomas in this study were completely resected (GTR), a GTR versus non-GTR comparison was not possible.

The following changes were made:

Table 2: P-values previously reported with negative exponents were updated to standard format (e.g., <0.001).

Table 2: A new row for “Convexity” was added, including the univariate p-value and odds ratio. This variable was not included in the multivariate analysis due to multicollinearity with IM location.

6: Large volume cut-off value should be included or the phrase large should change (suggestion: presence of IGC was associated with GTR (place OR, 95% CI, and P values), volume (place OR, 95%CI, and P values), and non-convexity (place OR, 95%CI, and P value).

We thank the Reviewer for the comment. The analysis was conducted using continuous volume increases, and we observed that each 1 cm³ increase in volume was associated with an increased risk of IGCs. To clarify this point, we have made the following changes:

Under results, manuscript page 8 line 215: “increasing IM volume (OR 1.1 per 1 cm3 increase, p<.001)

Under results manuscript page 9 line 250-251: This is likely related to larger IM volumes and a higher frequency of WHO grade 2 IMs in these patients (Table 2).

Congratulations on your line of research

We sincerely thank the Reviewer for thorough and insightful review, which has greatly helped us to improve the clarity and quality of our manuscript.

Reviewer #4

I have only 1 question:

Patients who underwent previous radiation therapy were excluded. However in 18 patients a WHO grade 2 meningioma was diagnosed. Did they undergo radiation treatment only after the first MRI at least one year later?

We thank the Reviewer for the valid question. In our study, we had 18 patients with grade 2 meningiomas, and all surgeries were complete resections. The operations took place between 2005 and 2018. These grade 2 meningiomas were on the less aggressive side of atypical meningiomas, so no patient received radiation therapy right after surgery. During follow-up after the study period, 3 patients needed radiation therapy because of radiological recurrence. The first patient had surgery in 2010 and received radiation in 2015. The second patient had surgery in 2016 and received radiation in 2019. The third patient had surgery in 2018 and received radiation in 2023.

And one textual remark: On p3, line 87/88, this sentence may improve when the authors write: ..., we assessed the occurrence of persistent FLAIR hyper intensity changes in IM patients without preoperative PTBE, who underwent gross total resection (GTR).

We thank the Reviewer for the good suggestion. We revised the manuscript as the Reviewer proposed:

Under introduction, Manuscript file page 3 line 86-88: “…,we assessed the occurrence of persistent FLAIR hyperintensity changes in IM patients who underwent gross total resection (GTR) without preoperative PTBE” � “…,we assessed the occurrence of persistent FLAIR hyperintensity changes in IM patients without preoperative PTBE who underwent gross total resection (GTR).”

---

## [Decision Letter · Decision Letter 1]

5 Dec 2025

Dear Dr. Mikael,

Thank you for submitting your manuscript to PLOS ONE. After careful consideration, we feel that it has merit but does not fully meet PLOS ONE’s publication criteria as it currently stands. Therefore, we invite you to submit a revised version of the manuscript that addresses the points raised during the review process.

We look forward to receiving your revised manuscript.

Kind regards,

Ryota Tamura

Academic Editor

PLOS One

Journal Requirements:

**Additional Editor Comments:**

Revision is needed again. Thank you very much for your effort.

Reviewers' comments:

Reviewer's Responses to Questions

**Comments to the Author**

Reviewer #3: All comments have been addressed

Reviewer #4: All comments have been addressed

2. Is the manuscript technically sound, and do the data support the conclusions?

Reviewer #3: Yes

Reviewer #4: Yes

3. Has the statistical analysis been performed appropriately and rigorously?

Reviewer #3: Yes

Reviewer #4: I Don't Know

4. Have the authors made all data underlying the findings in their manuscript fully available?

Reviewer #3: Yes

Reviewer #4: Yes

5. Is the manuscript presented in an intelligible fashion and written in standard English?

Reviewer #3: Yes

Reviewer #4: Yes

Reviewer #3: All comments were addressed; with these minor changes the article is suitable for publication in this highly recognized Journal.

Reviewer #4: I would favour adding one sentence about the radiation, just to avoid that readers think these iatrogenic changes may be the result of radiation. Otherwise, great paper.

**Do you want your identity to be public for this peer review?** For information about this choice, including consent withdrawal, please see our Privacy Policy

Reviewer #3: **Yes: ** Bernardo Cacho-Díaz, MD, MSc, PhD, FAAN

Reviewer #4: **Yes: ** Dennis R. Buis

---

## [Author Response · Author response to Decision Letter 2]

5 Dec 2025

Dear Dr Ryota Tamura,

We appreciate the reviewer’s thoughtful feedback and welcome the opportunity to provide a revised manuscript. The comments were carefully considered and have been fully addressed in the updated version. Our response is provided below. Thank you for your constructive input.

On behalf of all authors,

Reviewer #3

All comments were addressed; with these minor changes the article is suitable for publication in this highly recognized Journal.

Thank you for your continued review. We appreciate your initial comments and are pleased that our revisions have fully addressed them. We are grateful for your positive assessment of the manuscript.

Reviewer #4

I would favour adding one sentence about the radiation, just to avoid that readers think these iatrogenic changes may be the result of radiation. Otherwise, great paper.

Thank you for the helpful suggestion. We have added a clarifying sentence noting that none of the grade 2 meningiomas received radiation therapy before or during the follow-up period, thereby excluding radiation-related imaging effects. We appreciate your positive evaluation

Under Results manuscript page 8 line 212-214 added: “None of the WHO grade 2 meningiomas, nor any other meningiomas in the cohort, received radiation before or during the study’s observation period, ensuring that no imaging changes could be attributed to irradiation.”

---

## [Decision Letter · Decision Letter 2]

14 Dec 2025

Prevalence and associating factors for iatrogenic gliosis-like changes in surgically treated intracranial meningioma patients – A retrospective study of 255 meningioma patients

PONE-D-25-39472R2

Dear Dr. Mikael,

We’re pleased to inform you that your manuscript has been judged scientifically suitable for publication and will be formally accepted for publication once it meets all outstanding technical requirements.

Kind regards,

Ryota Tamura

Academic Editor

PLOS One

Additional Editor Comments (optional):

Reviewers' comments:

Reviewer's Responses to Questions

**Comments to the Author**

Reviewer #4: All comments have been addressed

2. Is the manuscript technically sound, and do the data support the conclusions?

Reviewer #4: Yes

3. Has the statistical analysis been performed appropriately and rigorously?

Reviewer #4: I Don't Know

4. Have the authors made all data underlying the findings in their manuscript fully available?

Reviewer #4: Yes

5. Is the manuscript presented in an intelligible fashion and written in standard English?

Reviewer #4: Yes

Reviewer #4: (No Response)

**Do you want your identity to be public for this peer review?** For information about this choice, including consent withdrawal, please see our Privacy Policy

Reviewer #4: **Yes: ** Dennis R. Buis

---

## [Editor Report · Acceptance letter]

PONE-D-25-39472R2

PLOS One

Dear Dr. Laajava,

I'm pleased to inform you that your manuscript has been deemed suitable for publication in PLOS One. Congratulations! Your manuscript is now being handed over to our production team.

Kind regards,

on behalf of

Dr. Ryota Tamura

Academic Editor

PLOS One